# FFA-IR: Towards an Explainable and Reliable Medical Report Generation Benchmark

**Mingjie Li[1†], Wenjia Cai[2†], Rui Liu[1], Yuetian Weng[1], Xiaoyun Zhao[1], Cong Wang[3], Xin Chen[2], Zhong Liu[2], Caineng Pan[2], Mengke Li[2], Yingfeng Zheng[2], Yizhi Liu[2], Flora D. Salim[4], Karin Verspoor[4], Xiaodan Liang[3], and Xiaojun Chang[4*]**

[1]Department of Data Science and Artificial Intelligence, Monash University
[2]State Key Laboratory of Ophthalmology, Zhongshan Ophthalmic Center, Sun Yat-Sen University
[3]School of ISE, Sun Yat-Sen University, Peng Cheng National Lab
[4]School of Computing Technologies, RMIT University

## Abstract

The automatic generation of long and coherent medical reports given medical images (*e.g.* Chest X-ray and Fundus Fluorescein Angiography (FFA)) has great potential to support clinical practice. Researchers have explored advanced methods from computer vision and natural language processing to incorporate medical domain knowledge for the generation of readable medical reports. However, existing medical report generation (MRG) benchmarks lack both explainable annotations and reliable evaluation tools, hindering the current research advances from two aspects: firstly, existing methods can only predict reports without accurate explanation, undermining the trustworthiness of the diagnostic methods; secondly, the comparison among the predicted reports from different MRG methods is unreliable using the evaluation metrics of natural-language generation (NLG). To address these issues, in this paper, we propose an explainable and reliable MRG benchmark based on **FFA I**mages and **R**eports (FFA-IR). Specifically, FFA-IR is **large**, with 10,790 reports along with 1,048,584 FFA images from clinical practice; it includes **explainable annotations**, based on a schema of 46 categories of lesions; and it is **bilingual**, providing both English and Chinese reports for each case. Besides using the widely used NLG metrics, we propose a set of nine human evaluation criteria to evaluate the generated reports. We envision FFA-IR as a testbed for explainable and reliable medical report generation. We also hope that it can broadly accelerate medical imaging research and facilitate interaction between the fields of medical imaging, computer vision, and natural language processing.

## 1   Introduction

Chest X-Ray, Lung CT-Scan, and Fundus Fluorescein Angiography (FFA) are among the most widely used imaging examinations in clinical practice. After images are captured, radiologists write a free-text description, or *report*, summarising observations and findings of lesions or abnormalities. This is done primarily to provide an interpretation of the images that supports making medical decisions. Given the complexity of image interpretation and to lighten the workload for radiologists considerably, researchers have proposed various deep neural networks (DNNs) based automatic medical report generation (MRG) methods [19, 20, 7]. Specifically, these methods employ convolutional neural networks (CNNs) [11, 27] to extract visual features, which are fed into recurrent neural networks (RNNs) [12, 28] to generate predicted reports. Although these methods have made some promising progress in the field of MRG, the black-box characteristics of DNNs discourage specialists

---

*Corresponding author. † Equal contribution

**(1) Clinical Practice**

Step.1 Classify the period of FFA images.  Step.2 Find the typical images.  Step.3 Diagnosis and write the report.

Prearterial  Venous
Arterial  Late
Arteriovenous

**Findings**:
1.左眼造影早期于黄斑区可见斑状强荧光，随造影时间延长，渐有轻微染料渗漏.
2.左眼造影期间黄斑区及周围可见散在斑状视网膜下出血性遮蔽荧光

**Impressions**:
中心凹下典型性CNVos；黄斑区及周围视网膜下出血性遮蔽荧光os

**Age**:
64

**Gender**:
Female

**(2) Reports Translation**

**Findings** :
1.左眼造影早期于黄斑区可见斑状强荧光，随造影时间延长，渐有轻微染料渗漏.
2.左眼造影期间黄斑区及周围可见散在斑状视网膜下出血性遮蔽荧光

Automatic translating

**Findings:**
1. In the early stage of left eye angiography, specular strong fluorescence was observed in the macular area, and slight dye leakage gradually occurred with the extension of angiography time.
2. Scattered subretinal hemorrhagic shadowing fluorescence was observed in and around the macular area of the left eye during angiography

Proofreading

**Findings:**
1. Early angiogram of the left eye demonstrating speckled hyperfluorescence in the macula. It develops slight leakage in the later stages of the angiogram.
2. Angiogram of the left eye demonstrating scattered speckled blocking affect from subretinal hemorrhage in and around the macula.

**(3) Explainable Annotation**

1. Early angiogram of the left eye demonstrating speckled hyperfluorescence in the macula. It develops slight leakage in the later stages of the angiogram.
2. Angiogram of the left eye demonstrating scattered speckled blocking affect from subretinal hemorrhage in and around the macula.

Figure 1: Process for creating FFA-IR. Firstly, we collect FFA images and reports from the clinical practice. To translate the reports, we invited bilingual ophthalmologists to proofread the automatically translated documents. They also labeled the described lesions along with FFA images and reports to provide explainable annotations.

and patients from trusting the predicted reports in clinical practice since medical decisions may have life-or-death consequences. To address this limitation, researchers have explored text-image attention mappings [7, 15] to explain the automatic generation procedure. However, the accuracy of these explanations is unclear. Since existing MRG datasets fail to provide explainable annotations, development of interpretable MRG methods to improve trustworthiness is a great challenge.

Besides explainable annotations, the lack of reliable evaluation tools hinders research advances. Natural-language generation (NLG) metrics, including BLEU [24], Cider [29], Meteor [4] and Rouge [22], have been widely used to evaluate the quality of the predicted reports. These methods focus on the linguistic similarity of target and source sentences and are based on counting the occurrences of overlapping N-grams, in which they treat each word in the sentences equally. They ignore the fact that certain words carry more weight in specific contexts. For image reports, identified lesions and their corresponding attribute descriptors are most important in diagnosis. Thus, these terms should carry larger weights in report quality evaluation than other words [34]. In addition, serious data bias commonly exists in medical reports. For example, a majority of the sentences in reports are descriptions of normal findings. In this context, overall performance in terms of standard NLG metrics will appear to be promising, although models are underfitting, particularly sentences describing abnormal findings and prone to repeat common sentences.

In this paper, we present a new benchmark, FFA-IR, towards an explainable and reliable MRG benchmark based on **FFA I**mages and **R**eports. There are two main motivations for building and releasing FFA-IR. In terms of clinical application, FFA is one of the most commonly used imaging methods for the diagnosis of retinal diseases [3]. Compared with other imaging methods, FFA can significantly improve the positive diagnosis rate. Thus, there is an urgent need to collect large-scale

FFA datasets with images and reports. In terms of scientific research, FFA-IR also provides a new challenge to MRG researchers. Compared against existing MRG datasets [16, 9], which provide only one or two views for each case, FFA-IR provides dozens of medical images for each case. Among these given medical images, only a few may capture lesions. In addition, the lesions are usually localized in a small area of the global image. Thus, we cannot simply concatenate the visual features from different views, as in traditional MRG methods [19, 34, 7] do, because other features will inundate lesion features in the same channel after concatenation.

The unique features of FFA-IR include:

- A large-scale medical dataset. Our FFA-IR contains $10,790$ reports describing $1,048,584$ FFA images in total, representing the most significant number of medical images among the existing medical report datasets. All these data are collected from real-world clinical practice and accurately represent the practical writing patterns of ophthalmologists.
- Explainable annotations. Compared with other datasets, our FFA-IR includes annotations of $46$ categories of lesions with a total of $12,166$ regions along with FFA images and reports to make the diagnosis process more explainable.
- Bilingual reports. The original reports obtained in the dataset are in Chinese. To make the dataset more broadly accessible, we also provide translations of these reports in English. The translations were derived from automatic translation followed by expert humans correction.

To provide solid and reliable benchmarks, we have conducted extensive experiments with both automatic and human evaluations. The primary experimental works are as follows: 1) We use NLG metrics to evaluate the quality of predicted reports from various approaches. 2) We conduct a detailed human evaluation by proposing a taxonomy of nine reliable criteria to quantitatively and qualitatively justify the predicted reports. 3) We calculate the Pearson Correlation [10] between the results of human and automatic evaluations to measure the clinical practicability of each NLG metric. 4) We calculate the Intersection-Over-Union (IOU) (also known as Jaccard similarity) between the lesion-image attention mapping regions and ground truth annotations to evaluate the accuracy of models' explanation. 5) We compare the quality of language reports generated by the same method in the two languages. 6) We investigate whether the predicted reports can be used for disease classification. Extensive analyses of these experiments can give helpful guidance for future research. We hope our FFA-IR can steadily attract and motivate more researchers to develop practical medical AI algorithms to support the work of ophthalmologists.

## 2 Related Work

**Medical report generation datasets.** Medical report generation tasks have attracted increasing attention from both artificial intelligence and clinical medicine fields. Many medical report datasets have been proposed, such as Open-IU [9], DEN [14] and COV-CTR [20]. We compare FFA-IR with nine widely-used and easily-accessed MRG datasets and report the statistics in Table 1. Open-IU and MIMIC-CXR [16] are the two widely-used medical report benchmarks, including chest X-Ray images and English written reports. PadChest [5] is another large-scale chest X-ray report dataset, which comprises $160,868$ images and multi-label annotated reports. MIMIC-CXR provides extra related disease impressions, which can be used for disease classification. Due to the characteristic of Chinese words, CX-CHR [21] and COV-CTR have a more considerable average report length than English medical report datasets. Compared with FFA-IR, DEN mainly contains Colour Fundus Photography (CFP) images ($13,898$ CFP and $1,811$ FFA). Table 1 shows that FFA-IR has the largest number of medical images and the longest average length of reports among all these datasets. There are also three more retinal datasets comprising retinal images and text. STARE [13] was conceived and initiated in 1975 and released in 2004 with 397 images including CFP and FFA. However, the texts provided with these images are short free-text diagnosis labels rather than observational reports of image findings and hence are unsuitable for training a medical report generation model. DIARETDB1 [17] is well annotated with lesion location and size yet has a limited number of CFP images. MESSIDOR [8] comprises $1,200$ CFP images and $600$ fine-gained French reports. Unlike all existing medical report datasets, FFA-IR provides explainable annotations by labeling 46 kinds of lesions in a total of $12,166$ regions along with FFA images and reports, which play an essential role in identifying disease and writing reports.

Table 1: Comparison of existing widely used MRG datasets, where * means the average number. Report length and number of lesions are marked as – for data sets that do not provide this figure.

| Dataset | Image | | | Report | | | Lesions |
|---|---|---|---|---|---|---|---|
| | Number | Modality | View* | Length* | Language | Cases | |
| Open-IU[9] | 7,470 | X-Ray | 2 | 32.5 | En | 2,955 | – |
| MIMIC-CXR[16] | 377,110 | X-Ray | 1 | 53.2 | EN | 276,778 | – |
| PadChest[5] | 160,868 | X-Ray | 2 | – | Es | 22,710 | – |
| CX-CHR[21] | 45,598 | X-Ray | 2 | 66.9 | Zh | 40,410 | 34 |
| COV-CHR[20] | 728 | CT-Scans | 1 | 77.3 | En/Zh | 728 | 2 |
| DEN[14] | 15,709 | CFP+FFA | 1 | 7 | En | | – |
| STARE[13] | 397 | CFP+FFA | 5 | – | En | 397 | – |
| DIARETDB1[17] | 89 | CFP | 1 | – | En | 89 | – |
| MESSIDOR[8] | 1,200 | CFP | 2 | – | Fr | 587 | – |
| FFA-IR | 1,048,584 | FFA | 87 | 91.2 | En/Zh | 10,790 | 46 |

**Data-driven medical report generation neural networks.** With these datasets, various MRG methods have been proposed. In the beginning, Jing *et al.* [15] presented an encoder-decoder framework and employed a co-attention mechanism over both visual and textual features to predict medical tags and generate a single sentence simultaneously. To generate multi-sentences, Xue *et al.* [33] fused multiple images modalities and adopted a topic level LSTM and a word-level LSTM [12] to generate multiple sentences. Li *et al.* [21] summarized a template base and combined retrieval-based and generation-based methods via reinforcement learning and directly selected sentences from their base. To utilize the medical domain knowledge, researchers started to integrate medical tag graphs into recurrent generative networks [34, 19]. With the success of Transformer [28] in NLP tasks, Li *et al.* [20] replaced the multi-level recurrent network with a medical tag graph Transformer to improve the long sequence processing efficiency and accuracy. Chen *et al.*[7] proposed a memory-driven Transformer to enhance the decoding procedure's memory. Despite progress in developing models, the lack of accurate explanation and reliable evaluation undermines the trustworthiness of these methods.

## 3 FFA-IR Dataset

To advance the medical report generation research and improve the conventional retinal disease treatment procedure, we release FFA-IR, a dataset focusing on diagnosing FFA images. For each case, we provide: 1) the clinically annotated Chinese reports and the translated English reports; 2) annotated lesion information, including lesion category and regions on FFA images, to explain the diagnostic procedure. We summarize our process for creating FFA-IR in Figure 1.

### 3.1 Motivation

The World Health Organization (WHO) estimates that 2.2 billion people have visual impairments, and 500 million of them are caused by specific retinal diseases such as age-related macular degeneration (AMD) and diabetic retinopathy (DR)[25]. FFA is one of the most common and essential examination methods in the differentiation, diagnosis, treatment, and prognosis of fundus ophthalmic diseases. FFA is a kind of dynamic imaging procedure, and as shown in Fig. 1, with sodium fluorescein flowing through the blood into the fundus vessels, the whole procedure can be divided into five parts: Preaterial, Arterial, Arteriovenous, Venous, and Late period. At different periods, ophthalmologists determine different diseases based on the morphology of different lesions. For example, the nature of new blood vessels in different areas from the fluorescein leakage pattern, the scope and size of the non-perfusion area of the retina. After browsing all the FFA images, ophthalmologists will select several typical FFA images according to their observations and write a report summarizing their findings. This process of reading and interpreting dozens of FFA images is laborious.

Compared with CFP imaging, FFA is a high-cost, invasive and complex imaging method but has a high confirmation rate. As some patients may be allergic to fluorescein, FFA is also not suitable for large-scale screening. Therefore, it is challenging and costly to collect large-scale data set FFA images and reports, making the FFA-IR collection highly valuable. A practical, interpretable, and

reliable MRG model derived using FFA-IR can assist ophthalmologists in understanding these images and improve the conventional retinal disease diagnosis procedure.

## 3.2   Data collection and Annotation

The data were collected from patients at the Zhongshan Ophthalmic Center of Sun Yat-Sen University in Guangzhou, China, during the period between 11/2016 and 12/2019. Institutional review board (No.2021KYPJ039) and ethics committee approval were obtained in Zhongshan Ophthalmic Center, Sun Yat-Sen University. This study followed the tenets of the Declaration of Helsinki [2]. All angiography images and reports were anonymized and de-identified before the analysis.

During the data collection period, our system captured $15,232$ reports, containing findings, impressions, and clinical information, along with $1,716,825$ DICOMs in which clinical information and pixel values of FFA images are stored. However, we removed some reports and FFA images due to data quality issues. First, there were some reports that could not be matched to FFA images with the same case ID number; Second, the pixel values were missing for some images when we converted the DICOMs to JPG pictures; Third, some reports were incomplete, with key information like findings or impressions missing. After processing the raw data, we finally obtained $10,790$ reports with $1,048,584$ FFA images for our FFA-IR data set.

**Annotator information** The original medical reports were generated by about 12 ophthalmologists of the fundus department. Around five ophthalmologists with 1-3 years of experience in fundus diseases generated reports under the supervision of residents or attending physicians in fundus specialty. About 3-4 residents or attending physicians in total, each with over five years of experience in the clinical retina, all of whom created reports independently. Finally, 3-4 senior retinal specialists with the title of professor or associate professor had been in the field of the retina for more than 15 years. They either wrote reports independently or helped make final decisions on complicated cases.

For image labeling and annotation, three ophthalmologists with about 2-5 years of experience in ophthalmology labeled lesion regions on FFA images. 2 residents with more than seven years in ophthalmology verified all lesion labels. One professor and one associate professor in ophthalmology checked the accuracy of the random sample of the labels and helped make final decisions on complicated cases. Images with problematical labels were discussed until all specialists agreed on the grading.

**Explainable annotations** Our FFA-IR annotation schema includes 46 categories of retinal lesions, such as Cystoid Macular Edema (CME) and Diabetic Macular Edema (DME). The schema was developed by the ophthalmologists based on their expert knowledge, and covers most typical retinal lesions. The schema can be viewed as defining the set of "explanations" that are relevant to the interpretation of the FFA images.

The ophthalmologists annotated each lesion with its minimum enclosing rectangle and providing the lesion category. All the lesions in one FFA are recorded in a dict format, and the key name is the combination of the case ID and the image name while the value is a list data, and each element contains the category and positional information.

The medical reports aim to describe the size, location, and period of detected lesions on the corresponding images. Therefore, any lesions annotated on the images should also be described in the report. The terms corresponding to each of the 46 categories of retinal lesions can be identified in the reports, and used to evaluate the accuracy of explanations generated by the models. Effectively, the schema serves as prior medical knowledge that enables connecting the visual features on the images and the linguistic information describing those features.

**Bilingual reports** To make the dataset more broadly accessible, we translate these reports to English and provide bilingual reports for each case. As it is laborious to translate tens of thousands of reports, we firstly uses DeepL Translator[31] to automatically translate all the reports and invited the bilingual ophthalmologists to proofread these reports. Due to the particularity of the Chinese language, we also provided a vocabulary containing medical nomenclature to help researchers tokenize the Chinese reports. Along with the bilingual reports, FFA-IR is the first benchmark to evaluate qualitative and quantitative influences of different languages on MRG methods. Thanks to these bilingual reports, FFA-IR can also facilitate the development of multi-modal machine translation models.

Table 2: The FFA-IR dataset statistics, where * represents the average number.

| | Attribute | Train | | Val | | Test | |
|---|---|---|---|---|---|---|---|
| | | En | Zh | En | Zh | En | Zh |
| Report | Length* | 63.4 | 91.3 | 63.6 | 91.1 | 63.5 | 91.0 |
| | Vocabulary(%) | 89.1 | 95.4 | 39.0 | 68.1 | 46.1 | 73.6 |
| Case | Number | | 8,016 | | 1,069 | | 1,604 |
| | Healthy(%) | | 5.6 | | 6.1 | | 5.5 |
| | Unhealthy(%) | | 94.4 | | 93.9 | | 94.5 |
| FFA | Image* | | 87.2 | | 87.3 | | 86.0 |
| Gender | Male(%) | | 55.6 | | 54.4 | | 57.8 |
| | Female(%) | | 44.4 | | 45.6 | | 42.2 |
| Eyes | Right(%) | | 29.0 | | 30.1 | | 29.7 |
| | Left(%) | | 39.6 | | 38.2 | | 40.1 |
| | Both(%) | | 31.4 | | 31.7 | | 30.2 |
| Age | Average | | 47.7 | | 47.6 | | 47.8 |
| | Range | | $3 \sim 92$ | | $3 \sim 87$ | | $4 \sim 91$ |
| Lesion | Number | | 9,336 | | 1,220 | | 1,610 |
| | Category | | 46 | | 46 | | 46 |

Table 3: The results of automatic and human evaluations, where B*N represents the N-gram of Bleu value, H*N represents the index of human evaluation, and T is the short for Transformer[28].

| | B1 | B2 | B3 | B4 | Meteor | Rouge | Cider | H1 | H2 | H3 | H4 | H5 | H6 | H7 | H8 | H9 |
|---|---|---|---|---|---|---|---|---|---|---|---|---|---|---|---|---|
| CoAtt[15] | 0.313 | 0.200 | 0.144 | 0.111 | 0.197 | 0.247 | 0.254 | 0.615 | 0.515 | 0.430 | 0.04 | 0.269 | 0.315 | **4.96** | **4.93** | **20.9** |
| Show-Tell[30] | 0.306 | 0.197 | 0.142 | 0.109 | 0.191 | 0.247 | 0.232 | 0.646 | 0.523 | 0.415 | 0.02 | 0.276 | 0.353 | **4.96** | **4.93** | 19.7 |
| Top-Down[1] | 0.320 | 0.217 | 0.162 | 0.127 | 0.207 | 0.289 | 0.363 | **0.684** | **0.584** | 0.430 | 0.01 | 0.292 | **0.376** | 4.83 | 4.70 | **20.9** |
| Gounded[35] | 0.396 | 0.319 | 0.261 | 0.218 | **0.229** | **0.353** | 0.576 | 0.538 | 0.361 | 0.423 | 0.03 | 0.307 | 0.292 | 4.82 | 4.84 | **20.9** |
| AdaAtt[23] | 0.292 | 0.181 | 0.127 | 0.095 | 0.205 | 0.236 | 0.234 | 0.553 | 0.338 | 0.515 | 0.06 | 0.384 | 0.284 | **4.96** | 4.78 | 18.6 |
| R2Gen[7] | 0.330 | 0.225 | 0.167 | 0.132 | 0.210 | 0.296 | 0.367 | 0.423 | 0.230 | 0.507 | **0.1** | 0.361 | 0.176 | 4.85 | 4.82 | 19.8 |
| CNN[11]+T | 0.321 | 0.211 | 0.154 | 0.122 | 0.198 | 0.268 | 0.283 | 0.423 | 0.238 | 0.523 | 0.079 | 0.369 | 0.176 | 4.77 | 4.76 | 18.9 |
| I3D[6]+T | 0.428 | 0.341 | 0.276 | 0.229 | 0.213 | 0.334 | 0.561 | 0.530 | 0.3 | 0.461 | 0.092 | 0.330 | 0.223 | 4.86 | 4.83 | 20.7 |
| F-R[26]+T | **0.443** | **0.355** | **0.288** | **0.240** | 0.205 | 0.341 | **0.590** | 0.590 | 0.3 | **0.576** | 0.084 | **0.392** | 0.215 | 4.83 | **4.93** | 18.4 |

## 3.3 Dataset statistics

We report the statistics of our FFA-IR in Table 2. In total, our FFA-IR dataset contains $10,790$ cases describing $1,048,584$ FFA images. For each case, FFA-IR provides FFA images, free-text reports, and explainable annotations.

Five percent of the cases in FFA-IR are entirely healthy and are negative training samples.[2] Consistent with most large-scale datasets for deep learning research, we created standard splits, separating the whole dataset into $75\%, 10\%, 15\%$, *i.e.*, $8,016$ (train), $1,069$ (val), and $1,604$ (test) cases, respectively. The vocabulary sizes of English and Chinese reports are $918$ and $6,181$, respectively. The training corpus covers most of the words, with words appearing less than 3 times in the corpus replaced by <unk> during the training process. Training Chinese models requires a larger wording embedding space which may influence the efficiency. Furthermore, there is no obvious data bias in the Gender and Age distributions. There are slightly more reports describing the left eye than the right in FFA-IR. The resolutions of FFA images in FFA-IR range from $384 \times 384$ to $3216 \times 2696$.

## 3.4 Data and code availability

Our dataset with all images and documentation, including bilingual reports, findings, explainable annotations, and lesion code dictionary, is hosted and maintained on PhysioNet under the following license: PhysioNet Credentialed Health Data License 1.5.0. It can be accessed at the following link: https://physionet.org/content/ffa-ir-medical-report/1.0.0/. Our start up codes can be accessed at https://github.com/mlii0117/FFA-IR, under the MIT licences.

---

[2]We note that this data set may therefore differ from data sets derived from diagnostic screening applications (such as breast cancer screening), where the positive samples would be expected to be in the minority.

Table 4: The results of IOU between lesion-image mapping regions and ground truth.

| | B4 | C | IoU |
|---|---|---|---|
| CoAtt[15] | 0.111 | 0.254 | 0.163 |
| R2Gen[7] | 0.132 | 0.367 | 0.203 |
| CNN[11]+T | 0.122 | 0.283 | 0.185 |
| F-R[26]+T | **0.240** | **0.590** | **0.312** |

Table 5: Classification results with using visual features and generated reports, where GT refers to the results using ground truth reports.

| | Vision | Report | GT |
|---|---|---|---|
| CoAtt[15] | 0.733 | 0.513 | |
| Top-Down[1] | 0.811 | **0.531** | |
| R2Gen[7] | 0.734 | 0.494 | 0.728 |
| I3d[6]+T | 0.762 | 0.494 | |
| F-R[26]+T | **0.821** | 0.527 | |

Table 6: Comparison of different language reports from the same models, where Zh, Zh-T and En represent generating reports by Chinese words, Chinese tokens, and English.

| | Bleu-4 | | | Cider | | | Hit |
|---|---|---|---|---|---|---|---|
| | Zh | Zh-T | En | Zh | Zh-T | En | |
| CoAtt[15] | 0.223 | 0.111 | 0.113 | 0.577 | 0.254 | 0.250 | En |
| R2Gen[7] | 0.231 | 0.132 | 0.131 | 0.623 | 0.367 | 0.367 | En |
| Gounded[35] | 0.303 | 0.218 | 0.220 | 0.854 | 0.576 | 0.569 | Zh-T |
| I3D[6]+T | 0.297 | 0.229 | **0.231** | 0.849 | 0.561 | 0.559 | En |
| F-R[26]+T | **0.365** | **0.240** | **0.231** | **0.882** | **0.590** | **0.590** | Zh-T |

# 4 Benchmarks

## 4.1 Approaches

In FFA-IR, we pose a new medical report generation task focusing on describing retinal diseases on FFA images. We present benchmark results over FFA-IR using baseline and existing MRG methods.

**Baseline model** MRG models usually contain two modules, visual extractor, and natural language decoder. In this paper, we develop three simple, transformer-based [28] baseline approaches, namely CNN [11]+Transformer [32], I3D [6]+Transformer and FasterRCNN [26]+Transformer. Firstly, we use ResNet [11], I3D, and Faster-RCNN as the visual extractor to extract spatial, temporal, and object features, respectively. For CNN+T, we employ ResNet [11] to extract the spatial features of each FFA image and then fuse them and feed 49 visual tokens to a transformer. For I3D+T, we first employ I3D pretrained on Kinetics [6] to extract temporal features and pad these features to 49 tokens and then feed to a transformer. For F-R+T, we first use Faster-RCNN pretrained with our lesion regions to extract object features. The object features have also been fused before being sent to a transformer. The batch size of training I3D+T is 32, while others are 2.

**Existing approaches** CoAtt[15], Show-Tell[30] and AdaAtt[23] propose similar CNN-LSTM neural network with different attention methods. Top-Down[1] and Gounded[35] extract object features as visual grounding for captions. R2Gen[7] integrate relational memory into Transformer to describe medical images with spatial features.

**Training details** We train these models using four NVIDIA GeForce GTX 2080Ti GPUs for 50 epochs and use the best models on the validation set for testing. Models are trained under cross entropy loss with ADAM[18] optimizer. Training time varies, taking $\sim$8 hours for I3D+T, $\sim$49 hours for spatial feature-based models, $\sim$62 hours for object feature-based models. The dimension of transformer-based models is 512, and the number of layers is 4. We set the initial learning rate to $5e-3$ and delay 0.1 every 10 epochs until $5e-5$. We use greedy decoding for inference models.

## 4.2 Evaluation metrics

**Report evaluation** We employ the automatic metrics and nine kinds of human evaluation results to evaluate the quality of the generated reports. The automatic metrics including BLEU [24], Cider [29], Meteor[4] and Rouge [22] aim to calculate the similarity between source and target sentence based on the occurrences of N-gram or word matching. However, these metrics cannot give reliable evaluations for medical fields as the detection of positive disease keywords should largely determine the quality

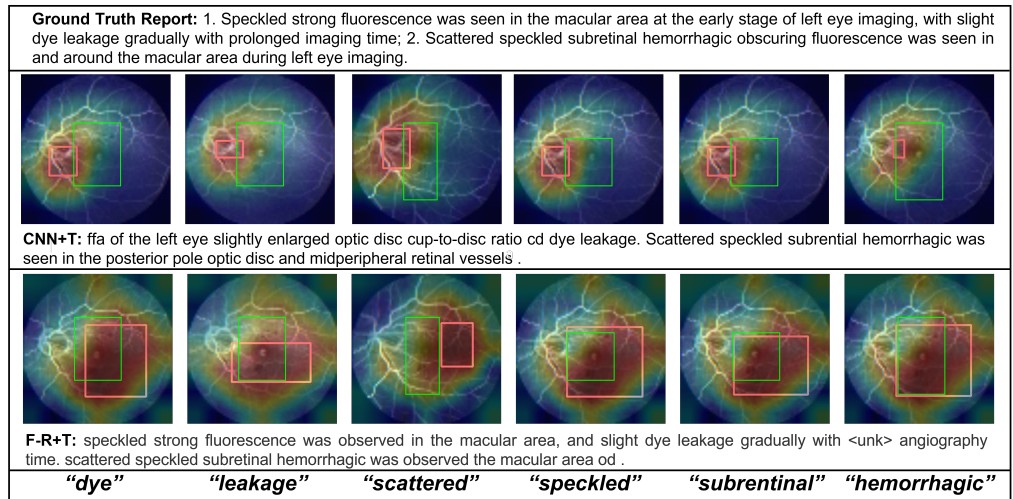

Figure 2: The visualization of lesion-image attention mapping regions and ground truth among samples from CNN-T and F-R+T, respectively, where the green boxes are the annotated region for each lesion word, and the red boxes are the lesion-image attention mapping regions.

of whole reports. Therefore, we propose a human evaluation of the reports, making use of four experienced ophthalmologists to answer a series of questions about the generated reports:

H1: Are the left and right eyes identified accurately?
H2: Is the imaging period accurately described?
H3: Does this report describe any lesion?
H4: Is the category of the described lesion accurate?
H5: Is the location of the described lesion accurate?
H6: Is the imaging period of the described lesion accurate?
H7: Fluency of the text, on a scale of [1-5] with 5 most fluent.
H8: Intelligibility of the text, on a scale of [1-5] with 5 most intelligible.
H9: The time savings (in seconds) achieved with the help of this report.

For binary questions H1-H6, we set "yes" for 1 and "no" for 0.

For H9, ophthalmologists first record the average time to diagnose one case with the first half samples. Then they will record the average time they used to diagnose one case with the help of generated reports with the remaining samples.

**Intersection-Over-Union** The existing methods visualize text-image attention mappings to explain the generation process. However, few of them justify the accuracy of their explanations due to a lack of ground truth regions. In FFA-IR, we quantify the accuracy of models' explanation by calculating the Intersection-Over-Union (IOU) (or Jaccard similarity) between the lesion-image attention mapping regions and ground truth regions. Then we draw the minimum rectangles to cover each maximum connection region. The IOU between generated rectangles and annotated regions is calculated. However, to capture the semantics of the annotation, each word in the relevant lesion label must correspond to a word that the model attends to in order to be counted.

**Mean average precision** Medical reports aim to describe lesions from the given medical images and can be considered as the interpretable foundations for disease diagnosis. Therefore, we conduct disease classification experiments and report the mean average precision (mAP) to compare the accuracy of each model.

## 4.3  Results and Analysis

**FFA-IR benchmark model** In Table 3, we report values of the automatic metrics and human evaluation to compare various models. Firstly, our FFA-IR benchmark model is F-R+T which achieves almost all the highest numbers of automatic metrics. It outperforms Gounded by 0.022, 0.014 in Bleu-4 and Cider, respectively.

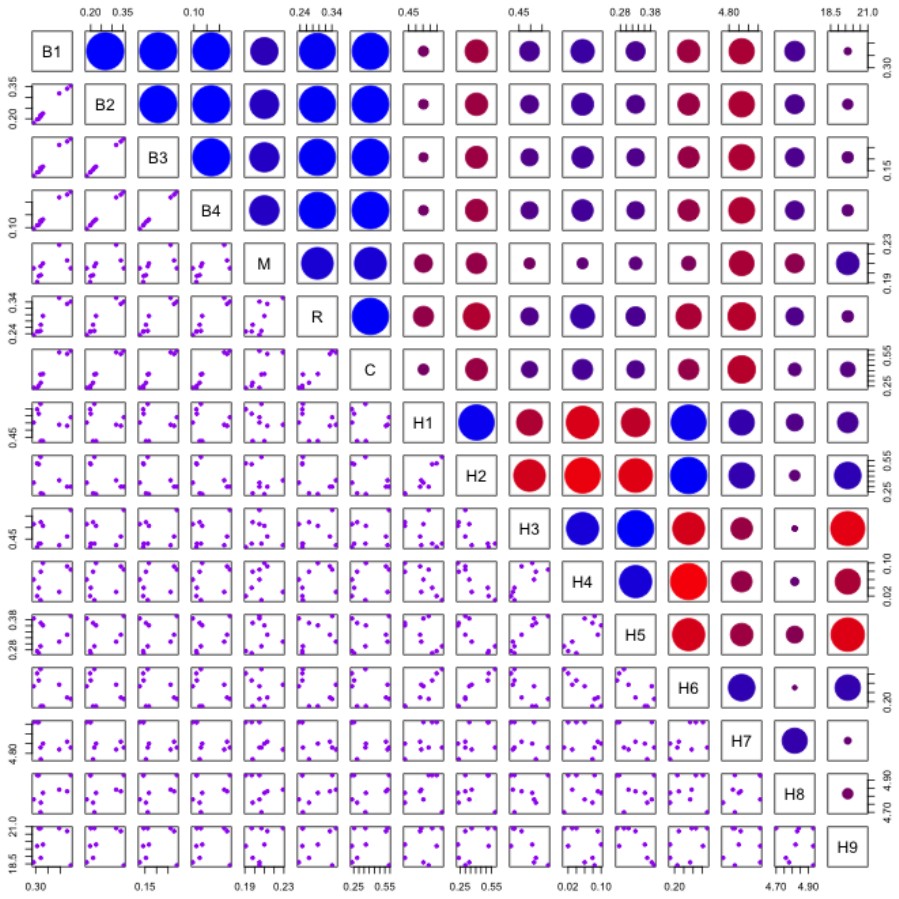

Figure 3: The Pearson correlations between each pair of metrics, where the blue and red refer to positive and negative correlation, respectively.

Although Gounded achieves stronger performance on the Meteor and Rouge metrics, we should note that both F-R+T and Gounded generate medical reports based on object features. Secondly, we find that in FFA-IR, the performance of models exploring object features is significantly higher than other models. These results demonstrate that lesion features are essential in MRG models, but they can be easily inundated by global features without supervised signals. Thirdly, in FFA-IR, Transformer is more efficient in generating long sequences than LSTM[12]. Fourthly, although CNN+LSTM models[15, 23, 30] perform poorly under the automatic metrics, ophthalmologists find that these models generate the most fluent and intelligible reports.

As mentioned, serious data bias exists in medical reports. CNN+LSTM models are prone to underfitting, generating repetitive and non-essential sentences. From the other human evaluation questions, we find that CNN+LSTM models have difficulty with accurate and detailed lesion information. Fifthly, based on H4 results, MRG models struggle to describe the correct category of lesions. Finally, to our knowledge, we are the first to verify the value of medical report generation for clinical diagnosis. Based on H9 results, the automatically generated reports can significantly save ophthalmologists time in the image interpretation and diagnosis procedure. Notably, the average time required for our ophthalmologists to diagnose cases in FFA-IR is $38.2$ seconds.

**Correlations between automatic metrics and human evaluation** Based on Figure 3, NLG metrics are correlated with each human evaluation criteria at various degrees. For instance, the Bleu values and H1-6 criteria are highly correlated with themselves and Meteor. Rouge is also negatively correlated with Bleu to a large degree. In addition, all B measures are correlated with H2-H8 with an absolute correlation of between $0.2 - 0.5$. Meteor, Rouge, and Cider are also correlated with human evaluation

criteria to a certain degree, especially with H6-H8, where the absolute correlation value range from 0.12 to 0.55. Another critical measure is the H7, as it is correlated with all other variables with an absolute correlation measure of bigger than 0.3. In sum, the existing NLG metrics are not the most reliable and appropriate evaluation tools in medical fields.

**Explanation accuracy** In Table 4, we calculate the IOU between the lesion-image attention mapping regions and annotated lesion regions to evaluate the accuracy of the explanation. We can find that F-R+T significantly outnumbers CNN+T, R2Gen, and CoAtt by 0.127, 0.109, and 0.149, respectively, proving that the FFA-IR benchmark model also has excellent explanation accuracy. We also visualize the explanation accuracy evaluation process in Figure 2. First, F-R+T can generate more fine-grained, coherent-semantic, and accurate reports than CNN+T. On the other hand, we can find that lesion-image attention mapping rectangles are closer to the ground truth regions.

**Disease classification results** Since the medical reports are used for facilitating the disease diagnosis procedure by ophthalmologists, we also conduct experiments to investigate whether the generated reports can be used for disease classification. Based on the results presented in Table 5, for each model, the results from using visual features alone are significantly higher than using the generated reports. Using ground truth reports, in contrast, can achieve comparable classification results. The results suggest that the generated reports are not yet strong enough to support disease classification.

**Does language affect the model?** In Table 6, we compare the quality of different language reports predicted from the same model. We find that different languages do not affect the model performance. However, the tokenization strategy does. Chinese sentences can be tokenized by words or tokens, as Chinese sometimes requires several words to describe a concept. The vocabulary sizes of Chinese words, Chinese tokens, and English words in FFAIR are 918, 2581, and 3241, respectively. Therefore, two reasons lead to that generating reports by Chinese words achieves higher automatic metric values. One reason is that generating reports by Chinese words has more matching words once the model recognizes a terminology; Another reason is that the word embedding space of Chines words is smaller than the other two's, decreasing the task difficulty. However, the human evaluation shows that these reports are dispreferred.

## 5  Conclusion and Limitations

This paper contributes a Fundus fluorescein Angiography Images and Reports (FFA-IR) dataset towards an explainable and reliable benchmark. The FFA-IR dataset has the following characteristics: 1) FFA-IR is a large-scale MRG dataset containing 10,790 reports along with 1,048,584 FFA images collected from clinical practice. 2) In FFA-IR, we label 12,166 lesion regions and collected reports and images to make the diagnosis procedure more explainable. 3) For each case, FFA-IR provides both English and Chinese reports that can facilitate medical multi-modal models. To the best of our knowledge, our work with FFA-IR is the first attempt to quantify the explanation of challenging medical report generation models, propose targeted human evaluation to judge the quality of predicted reports, and investigate the reliability of natural language generation metrics in the medical field. By releasing FFA-IR, we hope this task can be extensively explored in the future to advance research from both vision-and-language and medicine fields significantly and further improve the conventional retinal disease diagnosis procedures.

We have focused here on developing an explainable and reliable MRG model to describe lesions relating to retinal diseases identified in FFA images in Chinese or English language reports. Other usages of the dataset may include exploring the use of temporal information or interactions between related images to improve lesion detection or disease classification or to develop a multi-modal machine translation model by using medical images to align source and target sentences in the latent space.

However, our FFA-IR still has the following limitations: First, all these data are only collected from a single medical center. Second, as the original reports are collected from clinical practice, various writing patterns belonging to different report authors can be observed in FFA-IR, affecting the automatic metrics. Third, there are still several rare lesions that are not captured in FFA-IR. Fourth, FFA-IR suffers data bias due to the naturally unbalanced distribution of pathological statistics. Prior errors may also exist due to the unbalanced distributions across attributes, such as gender and age. Fifth, training models in FFA-IR require considerable GPU memories. The models have to read 87 images for each case on average.

## Acknowledgments and Disclosure of Funding

This work is partially supported by the Australian Research Council (ARC) Discovery Early Career Researcher Award (DECRA) under DE190100626, and the National Natural Science Foundation of China (82171034). We would like to acknowledge Prof.Feng Wen from Fundus Department, Zhongshan Ophthalmic Center, Sun Yat-Sen University for his data collection support. We would like to acknowledge the VoTT, Labelme, and Rect Label for providing us labelling platform. We would like to acknowledge DeepL for their automatic translation support.

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
