# OpenReview forum: "FFA-IR: Towards an Explainable and Reliable Medical Report Generation Benchmark "
_NeurIPS.cc/2021/Track/Datasets_and_Benchmarks/Round2 — NeurIPS 2021 Datasets and Benchmarks Track (Round 2)_

### Official Review · Reviewer_TARA · 2021-09-20
**Novel and robust dataset and benchmark for FFA that should inspire other medical imaging domains**

**Rating:** 9
**Confidence:** 3

**Strengths:**

- The authors do a great job of outlining the problem statement: (1) MRG methods are not explainable, but need to be for clinical use; (2) current NLG evaluation metrics do not translate well to the clinical report context.
- The authors present a novel dataset that is specific to FFA, but that could inspire similar such datasets in the medical imaging space. The dataset is huge (among the biggest for medical imaging datasets) and robust (it includes images, reports, annotations, and translations).
- The authors run a robust set of experiments to evaluate various models for report generation.
- The authors outline a clear motivation section for the dataset (3.1) that is very helpful context to provide to the reader.
- The authors are clear about how they source and build the dataset in Section 3.
- The authors do a great job to explaining the distribution of the dataset across various metrics.
- The authors propose a wonderful evaluation protocol that is specific to FFA evaluation but could serve as a great inspiration for other medical imaging datasets.

**Weaknesses:**

None of these are weaknesses, per se, but just some suggestions:
- It may be helpful to specify the years of experience and qualifications of the opthalmologist annotators.
- Small typos (very easily fixed!): line 40 ("to evaluate to"); line 133 ("It FFA"); line 321 ("different languages does"); line 341 ("usages of the dataset my")

**Additional Feedback:**

Overall, the paper is well written, the scope of the work is impressive, and I think this dataset and benchmark should serve as an inspiration for other medical imaging datasets.

**Clarity:**

- The paper is incredibly well written and clear, while still being thorough.
- Figure 1 was a very helpful visual to understanding the process of creating the dataset.

**Correctness:**

The claims made in the submission seem correct and any decisions the authors make seem thoroughly justified.

**Documentation:**

Yes.

**Ethics:**

The paper could include a few lines on the ethical use of this dataset.

**Relation To Prior Work:**

The authors include a robust section on related work, and very clearly, and helpfully, break it into two sections: one for medical report generation datasets and one for medical report generation models. The authors make it very clear how their dataset compares to existing datasets.

**Summary And Contributions:**

The authors present a large novel dataset FFA-IR that contains 10.7K reports describing 1M FFA images. The dataset also includes image annotations and translations of the reports from Chinese into English. The authors run a robust set of experiments to evaluate predicted reports, and also propose an evaluation protocol that is specific to clinical report generation.

---

> ### Author Response · Authors · 2021-09-29
> **To Reviewer TARA**
>
> Thank you for your recognition of our work. We have updated our paper following the suggestions.
> Firstly, we have specified the years of experience and qualifications of the ophthalmologist annotators in Section 3.2.
> Secondly, we have fixed all the typos.
> We hope that our FFAIR can broadly accelerate medical imaging research and facilitate interaction between the fields of automatic medicine and vision-and-language.

---

### Official Review · Reviewer_j1Xh · 2021-09-20
**New medical report generation dataset with extensive benchmark**

**Rating:** 8
**Confidence:** 3

**Strengths:**

This is overall a good paper.  The proposed dataset is very large considering that most medical imaging datasets contain data from less than 100 patients.  Despite the fact that evaluating the quality of generated reports is difficult, the authors use four automatically-generated metrics (namely BLEU, Rouge, Meteor, and Cider) as well as nine human evaluation criteria made by four experts.  The evaluation also includes IOU of predicted diseased regions.  I consider this dataset as a remarkable piece of work that will be of great utility for the AI and medical communities.

**Weaknesses:**

I do not see important weaknesses in that paper.  Unfortunately, the data comes from only one hospital which could lead to well-documented generalization issues.  Also, the dataset does not come with an online evaluation system that could centralize results obtained by various research teams in the future.  The dataset is also strongly oriented towards pathological cases (95% of the data contain a pathology) which could lead of important bias issues.

Another important missing aspect of this work is an inter-observer language variation evaluation.  Indeed, in the light of the results in table 6, it would have been nice to get the score obtained on reports written by experts (which i suspect would not be perfect in the light of these metrics) to see how far from being clinically usable these proposed methods are.

**Additional Feedback:**

Despite the fact that I am not an expert in medical report generation, I regard this paper as a very positive contribution, both for the AI and the medical community.  It is an impressive piece of work and the benchmark is convincing.   It is only too bad that this work does not come with an online evaluation system.  Overall, I believe this paper deserves to be accepted for publication.

**Clarity:**

This paper is very well written no problem.  Only detail : what is DOCAM?  Maybe the authors meant "DICOM"?

**Correctness:**

Overall, this work seams correct to me.  I downloaded the dataset and I could easily open images and read their associated annotations.  Also, the benchmark looks quite good with results from 5 different methods and several evaluation metrics, some being automatic while others being man made.  As mentioned before,  it would have been nice to get the score obtained on reports written by experts  to see how far from being clinically usable these proposed methods are.

**Documentation:**

The dataset comes will ample documentation on the data collection, patient distribution and license of use.  The dataset is hosted on physioNet where I have been able to download the data as well as its annotations.

**Ethics:**

There does not seam to be any ethical issues since the authors have had the approval of their ethic committee.

**Relation To Prior Work:**

The previous work section looks good to me.  But please consider the fact that I am not an expert in FFA nor in medical report generation...

**Summary And Contributions:**

The authors propose a new Fundus Fluorescein Angiograpy (FFA) dataset made to train and test medical report generation systems.  The dataset has data from 10,790 patients and more than 1 million images.  The dataset also contains annotations of 46 categories of lesions as well as bilingual Chinese-English reports.  The dataset (as well as the code that comes with it) has been rendered public.  Also, extensive benchmark has been made showing the effectiveness of these approaches.

---

> ### Author Response · Authors · 2021-09-29
> **To Reviewer j1Xh**
>
> Thanks for your encouraging and positive comments. We have released our code in our repository, and you are welcome to check them.
> We hope our FFAIR could benefit the community and encourage further research.
> We have also fixed the typos in the current version.
>
> In addition, we will propose an online evaluation tool since we realize that not every researcher has the resources to conduct the human evaluation.
> We plan to invite both junior and senior ophthalmologists as volunteers to judge the submitted results.

---

### Official Review · Reviewer_cE1y · 2021-09-22
**FFA-IR review**

**Rating:** 7
**Confidence:** 3
**Clarity:** The paper has a clear structure and i…

**Strengths:**

- given the big interest in medical data and the limited amount of data in this research area this is a valuable dataset
- the dataset can be used in various ways given the many different forms of annotations (reports in two languages, the lesion regions, etc.)

**Weaknesses:**

- the provided code repository is currently empty (except for some cryptic xml files).  (Update: Seems to be addressed now)
- It's known from similar fields that human annotators do frequently disagree with each other, which raises some concerns about the annotation procedure (Update: Seems to be addressed and clarified in the paper)

**Additional Feedback:**

Lines 161-164: You describe how you had two teams of annotators. It's known from similar fields that human annotators do frequently disagree with each other. Did you collect data on how often the validation team "corrected" the original annotations?
Lines 182-182: "tens of millions of reports" The dataset seems to have 10000 reports, still a lot of effort, but not "tens of millions"
Line 190: "text-free" = "free-text" ?!
Line 195: There seems to be a significant disparity between the Chinese and English vocabulary sizes. Is this due to the translation or inherent to the languages? In other words, would you expect originally English reports to have a larger vocabulary? And does this matter?
Line 319: If the features alone are already helpful (and seemingly outperform even human reports) while bothering to create reports at all?

**Correctness:**

The way the dataset was constructed seems sound. The newly proposed metrics are well motivated and compared to existing metrics.


**Documentation:**

The dataset is hosted on PhysioNet a repository for medical research data, this should guarantee reliable access. The paper describes the overall process of data collection in enough detail.  More information about how the data was annotated and if there were significant differences between the different annotators could be provided.

**Ethics:**

No. The limitations of the dataset are clearly stated, it seems to be balanced up to the fact that it relies on a single hospital as the source.

**Relation To Prior Work:**

Other datasets in this area are mentioned and the differences are explained.

**Summary And Contributions:**

The paper presents a new medical imaging dataset (Fundus Fluorescein Angiography). Besides containing a sequence of images for each case, the dataset contains Chinese and English reports and annotated lesion information. The lesion information contains the image region and the lesion's category. The dataset is an effort to provide explainable annotations to generate medical reports. In the second part, the paper runs experiments for several neural networks trained on this task. The networks' performances are evaluated both on established and newly proposed metrics.

---

> ### Author Response · Authors · 2021-09-29
> **To Reviewer cE1y**
>
> Thanks for your thoughtful comments.
>
> "The provided code repository is currently empty."
> We have released our code in our repository, and you are welcome to check them.
> We did not release the code prior to submission, since we need to come to an agreement with PhysioNet on the data format and the names of some directories.
> Therefore, we had to rewrite the data loader script to make sure our code fit the latest data.
> In comparison to the original dataset, we have made the following changes:
> 1. We transferred the annotation file from CSV to JSON format.
> 2. We renamed each directory and FFA image.
> 3. We also updated this information in our annotation files.
>
> "Two teams of annotators."
> In the latest revision, we provided more information in Section 3.2 to describe this procedure.
> We did not make statistics on how often the validation team corrected the annotation, since all the images with problematic labels were discussed thoroughly until all specialists reached an agreement on the grading.
>
> "Significant disparity between the Chinese and English vocabulary sizes."
> We reported the vocabulary sizes of Chinese words, Chinese tokens, and English words in the last paragraph of Section 4.3.
> These sizes are 918, 2581, and 3241, respectively.
> These differences are mainly due to the inherent nature of the languages.
> We cannot ensure that the English vocabulary size will always be larger than Chinese.
> In addition, the larger word embedding space will increase the task difficulty, and we have mentioned that it is another reason that predicted reports in Chinese words outperform others in NLG metrics.
>
> "Why bothering to create reports at all."
> The reports have value independent of the disease classification task as a human-readable interpretation of the images.
> These reports remain as the primary mechanism for disseminating conclusions from image analysis to clinicians.
> The results in Table.5 could only suggest that the predicted reports generated by current techniques were not strong enough to increase the automatic disease classification accuracy.
>
> We have also fixed the typos like "tens of millions" and "text-free."

---

### Comment · Area_Chair_M3To · 2021-09-28
**Discussion**

Thank you all for your reviews. Authors- please ensure to respond to any questions reviewers may have raised.

---

### Author Response · Authors · 2021-09-29
**For all reviewers**

Thank you for all the positive and encouraging comments.
We have made the following revisions in the latest version:
1. We have specified the years of experience and qualifications of the ophthalmologist annotators in Section 3.2.
2. We have provided more details in Section 3.2 to describe the annotation procedure. In summary, ……….
In summary, all the images with problematic labels were discussed thoroughly until all specialists reached an agreement on the grading.
3. We have reported the vocabulary sizes of Chinese words, Chinese tokens, and English words in the last paragraph of Section 4.3 and discussed the results in Table.5 in detail.
4. We have fixed all the typos.

Besides, we have released our code on our repository, and you are welcome to check them.

Lastly, we hope FFAIR could advance research from both vision-and-language and medical imaging understanding fields, and further improve the traditional retinal disease diagnosis procedures.

---

### Decision · Program_Chairs · 2021-10-09

**Decision:**

Accept

**Comment:**

Reviewers all agreed that this medical imaging dataset will be a plus to the community; it include both Chinese and English will help evaluating and training explainable annotations to generate medical reports. The scale of this dataset is big relative to other real-patient dataset. While evaluating quality of the reports remain challenging, authors show their best effort (eg BLUE, Rough) and human evaluations. Authors also responded to reviewers’ any remaining questions.